# Delayed room temperature phosphorescence enabled by phosphines

Guang Lu[1], Jing Tan[1], Hongxiang Wang[1], Yi Man[1], Shuo Chen[1], Jing Zhang[1], Chunbo Duan[1], Chunmiao Han [1] & Hui Xu [1] ✉

Organic ultralong room-temperature phosphorescence (RTP) usually emerges instantly and immediately decays after excitation removal. Here we report a new delayed RTP that is postponed by dozens of milliseconds after excitation removal and decays in two steps including an initial increase in intensity followed by subsequent decrease in intensity. The delayed RTP is achieved through introduction of phosphines into carbazole emitters. In contrast to the rapid energy transfer from single-molecular triplet states ($T_1$) to stabilized triplet states ($T_n^*$) of instant RTP systems, phosphine groups insert their intermediate states ($T_M$) between carbazole-originated $T_1$ and $T_n^*$ of carbazole-phosphine hybrids. In addition to markedly increasing emission lifetimes by ten folds, since $T_M \rightarrow T_n^*$ transition require >30 milliseconds, RTP is thereby postponed by dozens of milliseconds. The emission character of carbazole-phosphine hybrids can be used to reveal information through combining instant and delayed RTP, realizing multi-level time resolution for advanced information, biological and optoelectronic applications.

Organic ultralong room temperature phosphorescence (RTP) is a kind of afterglow from organic systems after excitation removal, which can persist several seconds and even hours[1]. With the merits of low cost and large-scale production, diverse applications of organic RTP materials were demonstrated, e.g. information encryption[2,3], anti-counterfeiting[4,5], biological imaging[6,7], sensing[8,9], and optoelectronic devices[10,11]. The rapid emergence of organic RTP materials in recent years is owing to the development of several approaches effectively overcoming the limitations of spin-forbidden triplet radiation and collision-induced triplet nonradiation by environmental factors (moisture, oxygen and temperature, etc)[12–14], including crystallization[15–17], self-assembly[18,19], H-aggregation[20–23], polymerization[24–26] and doping[27–33]. As a consequence, efficient and long-persist organic RTP in whole visible-light range were already demonstrated[34–37]. On the one hand, non-covalent interactions (e.g. π-π stacking[20] and intermolecular hydrogen bond) and rigid matrixes can prevent air infiltration and phonon vibration, therefore stabilize triplet excitons and suppress triplet nonradiation. On the other hand, besides incorporating heavy-atom effects of Br and I, aromatic heterocycles containing N, O, P and/or S atoms are also introduced to break through the El-Sayed rule[14] for increasing triplet population, because their $^1(n, \pi^*)$-dominant singlet excited states can markedly enhance spin-orbital coupling and intersystem crossing to $^3(\pi, \pi^*)$-featured triplet states, in contrast to $^1(\pi, \pi^*)$ states[38,39]. These structural designs were commonly combined to elongate lifetimes and improve quantum efficiencies.

It is noteworthy that all reported organic RTP phenomena are unidirectionally time correlated: after excitation, molecules transit to the first triplet states ($T_1$) from the first singlet excited states ($S_1$) through intersystem crossing, and then the excited energy is trapped by stabilized triplet states ($T_n^*$) for afterglow (Fig. 1a). The same $^3(\pi, \pi^*)$ characters of $T_1$ and $T_n^*$ leads to microsecond-level transition between them. So, for the human eye, this kind of RTP is instantaneous, and concurrent with fluorescence and phosphorescence, respectively, from the $S_1$ and $T_1$ states. In this case, after excitation removal, fluorescence and phosphorescence disappear, and RTP can be immediately observed by the human eye without any postponement. This kind of RTP phenomena can be defined as "instant" RTP, which has the

[1]Key Laboratory of Functional Inorganic Material Chemistry (Ministry of Education) & School of Chemistry and Material Science, Heilongjiang University, 74 Xuefu Road, 150080 Harbin, P. R. China. ✉e-mail: hxu@hlju.edu.cn

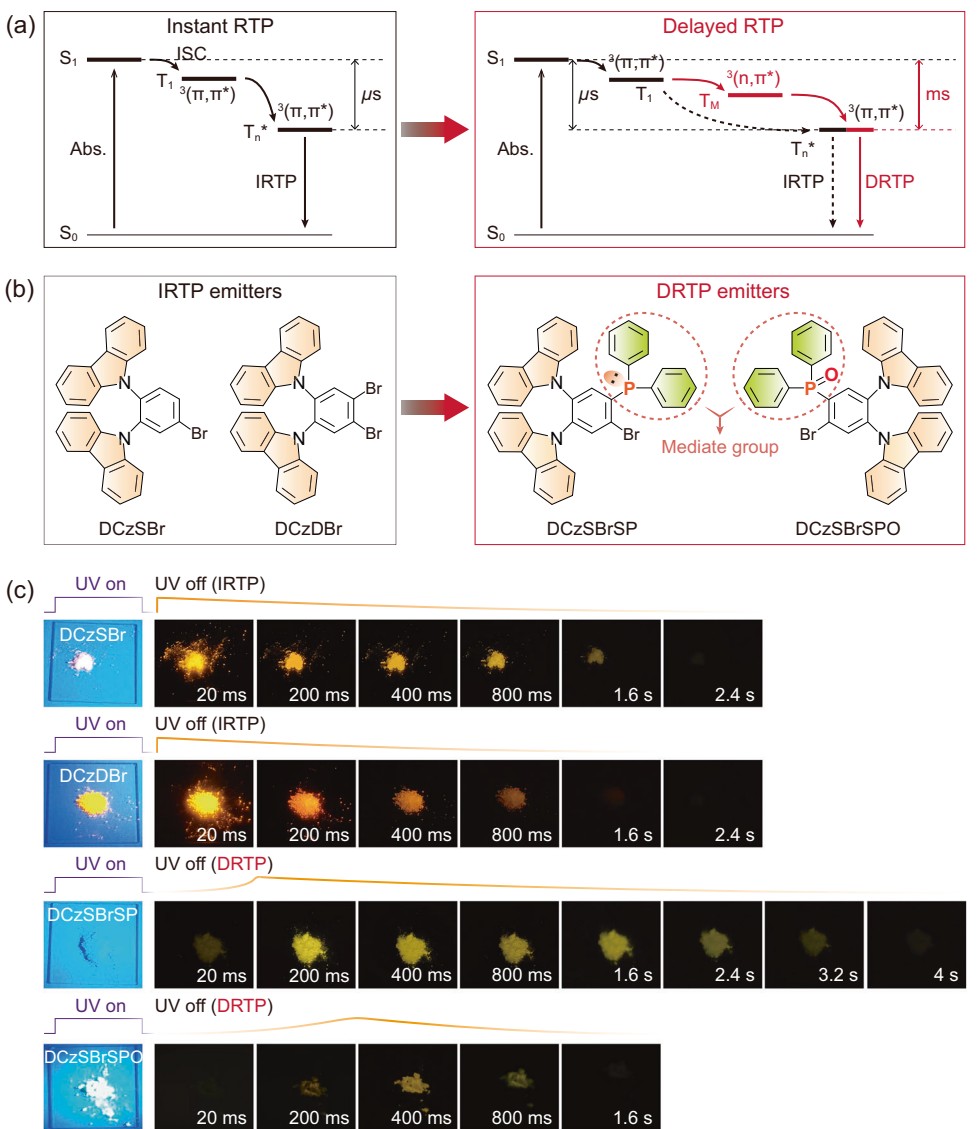

**Fig. 1 | Molecular design of binary phosphine-carbazole systems for delayed room temperature phosphorescence (DRTP). a** Illustrations of proposed energy transfer mechanisms for instant room temperature phosphorescence (IRTP) and delayed room temperature phosphorescence (DRTP). In contrast to rapid energy transfer in IRTP systems between excited states with constant π-π* characteristics, the incorporation of different n-π* featured intermediate energy levels increases transition probability, but elongates energy transfer routine and reduces transition rate, therefore postpones RTP emission. $T_1$ and $T_n^*$ refer to the first single-molecular and π-π stacking-stabilized triplet states localized on π-π* featured units; $T_M$ is the intermediate triplet state contributed by n-π* units. **b** Chemical structures of IRTP molecules DCzSBr and DCzDBr based on brominated carbazoles, and DRTP molecules DCzSBrSP and DCzSBrSPO containing phosphine units with n-π* characteristics as mediate groups. **c** Photos of DCzSBr, DCzDBr, DCzSBrSP and DCzSBrSPO powders excited with UV light at 365 nm and after UV excitation for 1.6-4.0 s.

intrinsic limitation in time resolution for naked eyes, regarding information hierarchy. Obviously, "delayed" RTP is "ideal" for realizing in-turn information display along timeline, which remains a big challenge in accurately modulating excited-state transitions. One feasible approach is inserting $^3$(n, π*)-featured triplet state as intermediate state ($T_M$) between $^3$(π, π*)-featured $T_1$ and $T_n^*$ state. In addition to the elongate energy transfer process, the different triplet characteristics could largely postpone transitions between, leading to delayed RTP. Nevertheless, despite incorporating $^3$(n, π*) state, instant RTP channel still exists, therefore competes with delayed RTP. In this sense, $^3$(n, π*)-featured mediate groups in the molecules are crucial.

The phosphorus atom has sufficient *n* electrons to enhance intersystem crossing[40,41]. Meanwhile, sp³ configuration of phosphine groups would modify intermolecular interactions, therefore modulate exited-state contributions of $^3$(n, π*) and $^3$(π, π*) components[41–45].

More importantly, we recently found that in *p*-carbazolylphenyl-diphenylphosphine, diphenylphosphine group provide $^3$(n, π*) energy level below $^3$(π, π*) energy level of carbazole group, rendering twofold increased RTP lifetime[46]. It can be expected that energy level relationship of carbazole-phosphine hybrids can be further accurately optimized to postpone $T_n^*$ population by dozens of milliseconds, reaching the time limit can be recognized by the human eyes.

As a proof of concept, we construct two carbazole and bromine-substituted phosphine derivatives named DCzSBrSP and DCzSBrSPO (Fig. 1b and Supplementary Note 1, Supplementary Fig. 1). Compared to brominated bicarbazole analogs DCzSBr and DCzDBr with instant RTP, whose two carbazoles at *ortho* position enhance the stability of $^3$(π, π*) states, diphenylphosphine and diphenylphosphine oxide of DCzSBrSP and DCzSBrSPO are introduced as $^3$(n, π*)-featured mediate groups. The heavy atom effect of bromines is utilized to further

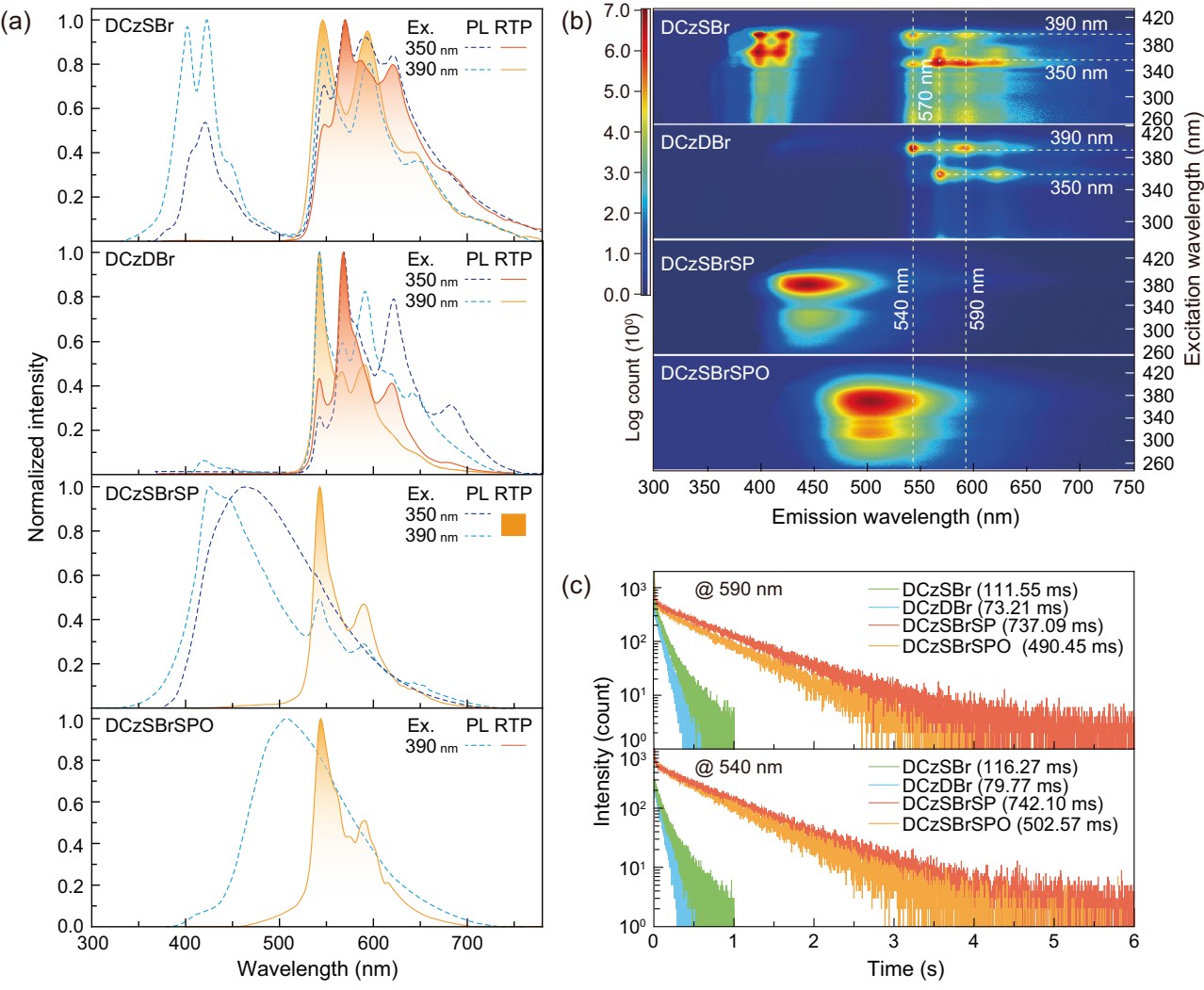

**Fig. 2 | Photophysical properties of RTP molecules. a** Steady-state photo-luminescence (PL, dashed lines) and time-resolved RTP (solid lines) spectra of DCzSBr, DCzDBr, DCzSBrSP and DCzSBrSPO powders under ambient condition. RTP spectra were recorded after a delay of 30 ms. For DCzSBr, DCzDBr and DCzSBrSP, two excitation wavelengths of 350 and 390 nm are chosen to indicate the excitation dependence of their PL and RTP emissions. **b** Excitation-emission mapping of DCzSBr, DCzDBr, DCzSBrSP and DCzSBrSPO powders under ambient conditions. **c** Time decays of emission peaks at 590 (above) and 540 nm (below) for DCzSBr, DCzDBr, DCzSBrSP and DCzSBrSPO powders excited by 350 nm.

improve intersystem crossing and triplet radiation. Under UV excitation at 365 nm, the emissions from the powders are white and yellow for DCzSBr and DCzDBr, and greenish blue for DCzSBrSP and DCzSBrSPO (Fig. 1c). After excitation removal, DCzSBr and DCzDBr immediately exhibit orange RTP with durations of ~2 s. In contrast, yellow RTP from DCzSBrSP can still be recognized at ~8 s. More importantly, RTP intensity of DCzSBrSP gradually increase during the initial 200 ms and then decrease, indicating the desired "delayed" RTP. DCzSBrSPO also reveals a similar delayed RTP phenomenon with doubled postponement time (400 ms). Notably, instant RTP of DCzSBrSP and DCzSBrSPO can also be recognized, but largely weaker than their delayed RTP and instant RTP of DCzSBr and DCzDBr. It indicates competition between instant and delayed RTP and the predominance of $^3(n, \pi^*)$ states in triplet transitions of DCzSBrSP and DCzSBrSPO.

## Results

Steady-state photoluminescence spectra of DCzSBr powder consist of two bands respectively centered at 420 and 600 nm, in which the yellow band is nearly the same with its time-resolved RTP spectra; while photoluminescence and RTP spectra of DCzDBr are nearly

overlapped, since its triplet radiation is facilitated by doubled atom effect of its two bromine atoms (Fig. 2a and Supplementary Table 1). In contrast, RTP components in steady-state photoluminescence spectra of DCzSBrSP and DCzSBrSPO are markedly weaker, giving rise to their emission peaks at 465 and 502 nm, respectively. Nonetheless, their time-resolve RTP spectra correspond to yellow emissions with main and shoulder peaks at ~540 and ~590 nm, respectively. RTP spectra of DCzSBrSP and DCzSBrSPO are the same when excited at 350 and 390 nm, but DCzSBr and DCzDBr excited with 350 nm reveal two additional RTP peaks at ~570 and ~620 nm. Excitation-emission spectral mapping further demonstrates that when excitation wavelength <390 nm, RTP spectra of DCzSBr and DCzDBr are unchanged, but which are different to their RTP spectra excited with 390 nm (Fig. 2b). Furthermore, all these four materials display nearly identical RTP spectra excited by 390 nm, corresponding to the same transition processes of their carbazole-originated $T_n^*$ states. In opposite to DCzSBr and DCzDBr, RTP spectra of DCzSBrSP and DCzSBrSPO are independent on excitation wavelengths, despite their electronic spectra identical to those of DCzSBr and DCzDBr (Supplementary Note 2, Supplementary Fig. 2). It means phosphine groups of DCzSBrSP and DCzSBrSPO indeed

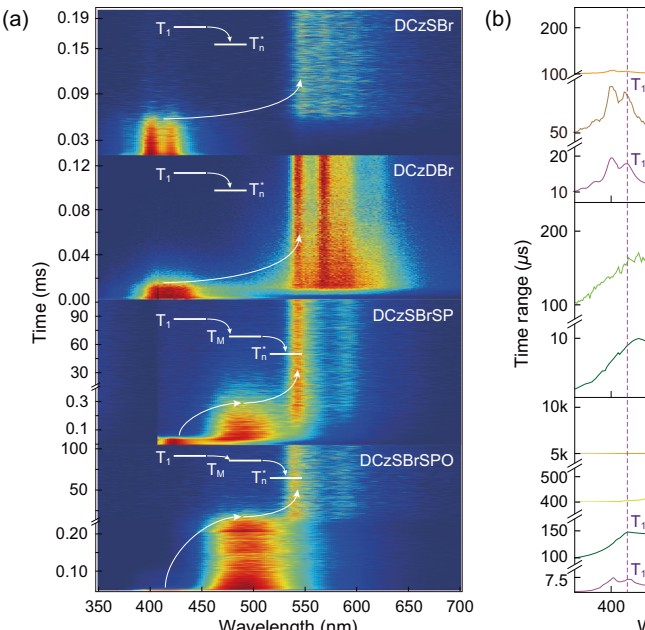

**Fig. 3 | Delayed RTP mechanism of carbazole-phosphine hybrids. a** Sliced time-resolved emission spectra (TRES) of DCzSBr, DCzDBr, DCzSBrSP and DCzSBrSPO powders. The corresponding energy transfer processes are highlighted with arrows and insets. For DCzSBr and DCzDBr, only direct energy transfer from the $T_1$ to $T_n^*$ states can be recognized, corresponding to typical IRTP process; while, for DCzSBrSP and DCzSBrSPO, new phosphorescence bands between their $T_1$ to $T_n^*$ states are observed and attributed to the $T_M$ states, which transfer triplet energy

from the $T_1$ to $T_n^*$ states. **b** Emission spectral evolution of DCzSBrSP at four representative stages of 5–10 μs, 100–150 μs, 400–500 μs and 5–10 ms. Sliced TRES profiles of DCzSBr and TPPBr powders are included for comparison. TPPBr with the structure same with phosphine mediate group in DCzSBrSP is used to clarify origin of intermediate energy levels ($T_M$). The correspondence between the spectra of DCzSBrSP, and DCzSBr and TPPBr are indicated with the same curve colors and dash lines.

provide $T_M$ energy levels to facilitate energy transfer to low-lying $T_n^*$ states.

For RTP peaks at 540 nm excited by 390 nm, compared to DCzSBr and DCzDBr with RTP lifetimes of 116 and 80 ms, DCzSBrSP exhibits the longest RTP lifetime at 540 nm reaching 742 ms, which is elongated by more than 6 and 9 folds, respectively (Fig. 2c). Despite shorter than that of DCzSBrSP, RTP lifetime of DCzSBrSPO also increases to 503 ms. Furthermore, RTP lifetimes at 590 nm are nearly equal to those at 540 nm for all these four materials, indicating consistent photophysical procedures. However, sulfide of DCzSBrSP does not exhibit long-persistence room temperature phosphorescence, since the lifetimes of its powder are less than 20 ms (Supplementary Figs. 3 and 4). The comparable lifetimes of DCzSBrSP and DCzSBrSPO imply their similar RTP mechanism, which is undoubtedly different to that for DCzSBr and DCzDBr. Obviously, this mechanism difference also renders the unique delayed RTP for DCzSBrSP and DCzSBrSPO.

Carbazole is the chromophore of all these four materials, which makes the main contributions to single-molecular excited-state characteristics, therefore, their electronic absorption, photoluminescence and phosphorescence spectra in dilute solutions ($10^{-5}$ mol L$^{-1}$ in dichloromethane) are nearly identical (Supplementary Fig. 5 and Supplementary Table 2). The lifetimes of the solutions are similar to the fluorescence lifetimes of the powders, indicating the initial states of emissions from the powders are their single-molecular $S_1$ states (Supplementary Figs. 6, 7). The details of transition processes for RTP from these four materials are investigated with time-resolved emission spectra (TRES) of DCzSBr, DCzDBr, DCzSBrSP and DCzSBrSPO powders, which indicate the step-by-step evolution from the $T_1$ states to $T_n^*$ states (Fig. 3a and Supplementary Fig. 8). It is shown that in the time range from microsecond to millisecond, all these four materials firstly revealed the microsecond-scaled emissions centered at ~420 nm, which are nearly identical to single-molecular phosphorescence spectra originated from the $T_1$ states (insets of Supplementary Figs. 5,

9, 10). Without any intermediate processes, $T_1$ states of DCzSBr and DCzDBr transit to the $T_n^*$ states within 60 and 20 μs, respectively. Such short time gaps can not be recognized by human eyes, giving rise to instant RTP. On the contrary, only a small proportion of the $T_1$ states for DCzSBrSP and DCzSBrSPO immediately transit to their $T_n^*$ states after dozens of microseconds, rendering negligible instant RTP. But, the majority of their $T_1$ states evolve to the $T_M$ states, which are embodied as the additional triplet bands at ~490 nm between single-molecular phosphorescence and RTP, and finally transit to the $T_n^*$ states of DCzSBrSP and DCzSBrSPO after 30 and 50 ms, respectively, resulting in delayed RTP at 540 nm. Consequently, "dark time" of dozens of milliseconds from excitation removal to delayed RTP is long enough to be recognizable by naked eyes.

Single-crystal X-ray diffraction data indicate that π-π stackings between carbazole groups of adjacent DCzSBrSP and DCzSBrSPO molecules are similar to those of DCzSBr and DCzDBr, but DCzSBrSP and DCzSBrSPO reveal additional intermolecular $p$-π interactions between adjacent carbazole and phosphine groups (Supplementary Note 3, Supplementary Figs. 11–15). Quantum chemical calculation shows that at triplet excited state, the frontier molecular orbital distributions and energy levels of π-π stacked DCzSBrSP and DCzSBrSPO dimers are similar to those of DCzSBr and DCzDBr dimers; while, DCzSBrSP and DCzSBrSPO dimers with $p$-π interactions reveal the increased contributions of diphenylphosphine groups to triplet excited states, rendering the higher triplet energy levels than the π-π stacked dimers (Supplementary Note 4, Supplementary Figs. 16–20).

DCzSBrSP is considered as the combination of DCzSBr and (2-bromophenyl)-diphenylphosphine (TPPBr) as RTP and delay units, respectively (Fig. 3b). The sliced time-resolved emission spectra of DCzSBrSP illustrate the excited-state evolution of DCzSBrSP after excitation removal: (i) during <7.5 μs, phosphorescence spectrum centered at ~420 nm corresponds to carbazole-originated $^3(π, π^*)$ states identical to that of DCzSBr at 10–20 μs; (ii) during 7.5–150 μs,

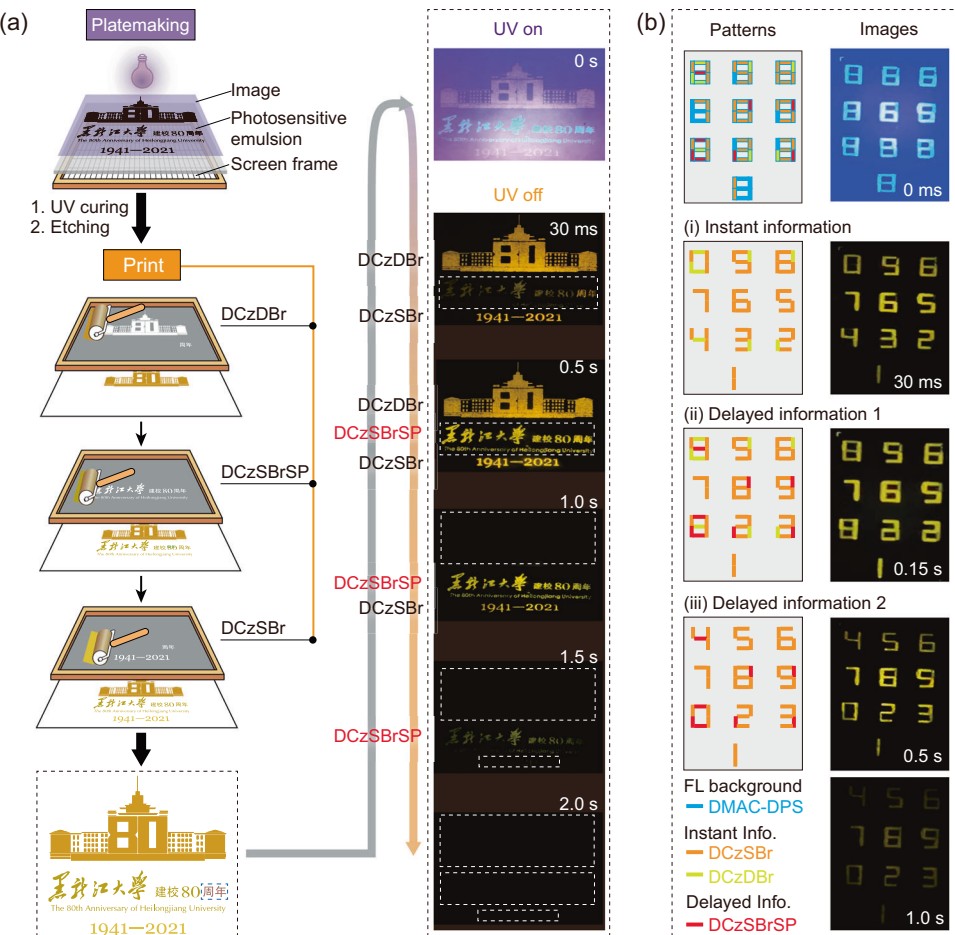

**Fig. 4 | Incremental afterglow display based on DRTP. a** Screen printing process of anti-counterfeiting pattern containing the information of the 80th anniversary for Heilongjiang University (left), and the changing process of emission patterns under 365 nm excitation at 0 s and at several representative stages at 30 ms and 0.5–2.0 s after UV turn-off (right). DCzSBr, DCzDBr and DCzSBrSP were used to prepare the pattern, which are indicated at the left of their corresponding patterns. The emerging of DCzSBrSP based Chinese characters can be distinguished between 30 ms and 0.5 s. **b** Nonlinear time-encoded security application exampled with a password panel based on DCzDBr, DCzSBr and DCzSBrSP. A thermally activated delayed fluorescence (TADF) molecular DMAC-DPS with microsecond-level lifetime was used to initiate emission pattern under UV excitation. After UV turn-off, three different RTP patterns can be distinguished: (i) instant information at 30 ms formed by IRTP of DCzSBr and DCzDBr; (ii) delayed information 1 at 0.15 s formed by IRTP of DCzSBr and DCzDBr and DRTP of DCzSBrSP; (iii) delayed information 2 after 0.5 s formed by IRTP of DCzSBr and DRTP of DCzSBrSP.

along with 420-nm phosphorescence decreases, a new phosphorescence band centered at ~490 nm increases, which is consistent with $^3(n, \pi^*)$-featured phosphorescence spectrum of TPPBr; (iii) during 400−500 µs, carbazole-based 420-nm phosphorescence disappears, but accompanied by 490-nm band, another new band peaked at 540 and 590 nm arises, which is identical to RTP spectrum of DCzSBr from the $T_n^*$ state; (iv) at >5 ms, only RTP band remains.

Therefore, it is rational that the transition process of DCzSBrSP and DCzSBrSPO is $^3(\pi, \pi^*)$ state of carbazole group → $^3(n, \pi^*)$ state of phosphine group → $^3(\pi, \pi^*)$ featured $T_n^*$ state of carbazole group → RTP. The transition from $^3(n, \pi^*)$ state to $T_n^*$ state is the slowest step at millisecond scale, resulting in the special delayed RTP. In this case, phosphine moieties play the key role as mediate groups, which provide intermediate $^3(n, \pi^*)$ states as $T_M$ states to trap triplet energy for dozens of milliseconds.

DCzSBrSP with delayed RTP and DCzSBr and DCzDBr with instant RTP were adopted to realize incremental information display by utilizing their different occurrence and decay times (Fig. 4). The ground powders of DCzDBr, DCzSBrSP and DCzSBr were used as solid inks for screen-printing three elements of main-building pattern, Chinese and English characters and numbers in a logo for 80th anniversary of Heilongjiang University, respectively (Fig. 4a). After UV turn-off, afterglows of main-building pattern and numbers firstly appeared, and then the characters as information increment became visible and the brightest at 30 and 500 ms, respectively (Movie S1). Subsequently, in accordance with the RTP lifetimes of DCzDBr, DCzSBr and DCzSBrSP, main-building pattern, numbers and characters successively disappeared within 2 s.

This combination of instant and delayed RTP can easily realize multi-level and time-resolved anti-counterfeiting and information encryption. A set of numbers "8987898aa1" as a triplet-encryption password was also prepared with DCzSBr, DCzDBr and DCzSBrSP inks (Fig. 4b and Supplementary Note 5, Supplementary Fig. 21). Bis[4-(9,9-dimethyl-9,10-dihydroacridine)phenyl]sulfone (DMAC-DPS) with bright blue fluorescence was used as background to cover the different emissions of the inks and form initial numbers of "8888888888" under UV excitation. At 30 ms after UV off, since DCzSBrSP based bars were non-emissive, instant RTP of DCzSBr and DCzDBr-based bars generated an instant information of "0987654321" (Movie S2). Then, at 150 ms, RTP of DCzSBrSP-based bars became comparable, giving rise to the first delayed information same as the correct password. And then, DCzDBr based bars with the shorted lifetime disappeared at 500 ms to generate the second delayed information of ""4567890231". Obviously, this triple encryption was mainly dependent on delayed

RTP, which added new information at the initial stage of emission decay, thereby doubled information depth. Furthermore, time windows for recognizing encrypted information, e.g. instant information herein, is only dozens of milliseconds, which is delicately balanced for purposeful recognition, but makes unintentional cracking far more difficult.

## Discussion

In conclusion, we have demonstrated delayed RTP in carbazole-phosphine hybrid organic molecules featuring mixed π-π and $p$-π intermolecular interactions in solid states. Different to common instant RTP immediately occurred after excitation removal, delayed RTP can only be recognized with a postponement of dozens of milliseconds, and then gradually enhanced in the next hundreds of milliseconds. Therefore, delayed RTP can realize incremental information display, rather than common unidirectional decay. Photophysical results suggest that phosphine groups provide the $^3(n, \pi^*)$-featured intermediate state ($T_M$) between $^3(\pi, \pi^*)$-featured $T_1$ and $T_n^*$ states to form a step-by-step transition process. The different excited-state characteristics postpone $T_M \rightarrow T_n^*$ transition by dozens of milliseconds, resulting in delayed RTP. This work not only presents a fundamental for controllably modulating transition process in delayed RTP systems containing π-π and $p$-π segments, but also supports a flexible and unique platform for multiple time-resolution optical applications in bio-imaging, passive programmable display, multi-level anti-counterfeiting, and so on.

## Methods

### General information

The crystals suitable for single-crystal XRD analysis were obtained through vapor-phase diffusing $n$-hexane to dichloromethane solution (5 ml) of the materials (10 mg). All diffraction data were collected at 295 K on a Rigaku Xcalibur E diffractometer with graphite monochromatized Mo Kα ($\lambda = 0.71073$ Å) radiation in ω scan mode. All structures were solved by direct method and difference Fourier syntheses. Non-hydrogen atoms were refined by full-matrix least-squares techniques on F2 with anisotropic thermal parameters. The hydrogen atoms attached to carbons were placed in calculated positions with C−H = 0.93 Å and U(H) = 1.2Ueq(C) in the riding model approximation. All calculations were carried out with the SHELXL97 program. Absorption spectra were measured using a SHIMADZU UV-3150 spectrophotometer. Photoluminescence spectra were measured with an Edinburgh FPLS 1000 fluorescence spectrophotometer. The time decay spectra was measured using Time-Correlated Single Photon Counting (TCSPC) method with a picosecond hydrogen lamp for 100 ps−10 $\mu$s and a microsecond pulsed Xenon light source for 1 μs-10 s lifetime measurement, the synchronization photomultiplier for signal collection and the Multi-Channel Scaling Mode of the PCS900 fast counter PC plug-in card for data processing. Lifetime values were simulated by a single exponential fitting function in Fluoracle software. For anti-counterfeiting and encryption applications, a circular ultraviolet flashlight (GET-104) was used as the excitation source with a power of 3 W and the peak wavelength at 365 nm, whose spot diameter is about 2 cm at the distance of 10 cm above the samples.

### General procedure for pattern preparation

(i) Platemaking: screens used for printing were customized according to the separated elements of the patterns (Fig. 4a and Supplementary Note 5, Supplementary Fig. 21).

(ii) Ink preparation: all the materials were ground and then sieved. The inks were prepared through uniformly dispersing the powders in aloe vera gel. Inks of DCzDBr, DCzSBrSP and DCzSBr were directly used to print the patterns in Fig. 4a; while, for Supplementary Fig. 21 and Fig. 4b, Inks of DCzDBr, DCzSBrSP and DCzSBr were mixed with

DMAC-DPS based ink, so that under UV excitation, RTP can be covered by blue fluorescence from DMAC-DPS.

(iii) Printing: The separated elements of the patterns were sequentially printed with the corresponding inks through screens prepared in step (1). For the pattern in Fig. 4a, DCzDBr-based ink was firstly used to print the "main building of Heilongjiang University" on filter paper. After this pattern dried, DCzSBrSP based ink was used to print the university name in Chinese and English. Finally, DCzSBr-based ink was used to print the year numbers. The Chinese characters of "周年" were used to calibrate the positions of all the elements. For the patterns in Fig. 4b, as shown in Supplementary Fig. 21, four screens were used to the patterns on the filter paper through four steps and the corresponding inks.

### Sample preparation

**Synthesis of 9,9'-(4-Bromo-1,2-phenylene)bis(9H-carbazole) (DCzSBr).** A mixture of carbazole (4.01 g, 24 mmol), 4-bromo-1,2-difluorobenzene (1.93 g, 10 mmol) and KOH (1.35 g, 24 mmol) in dimethyl sulfoxide (DMSO) (45 mL) was stirred at 140 °C for 1.5 h under argon atmosphere. After cooling to room temperature, the mixture was poured into water and then filtered. The crude product was dried and then purified by column chromatography and recrystallization from methanol and ethyl acetate, affording white solid with a yield of 72% (3.51 g). $^1$H NMR (TMS, CDCl$_3$, 400 MHz): $\delta = 7.983$ (d, $J = 2.0$ Hz, 1H), 7.766−7.833 (m, 5H), 7.702 (d, $J = 8.4$ Hz, 1H), 7.109−7.175 (m, 4H), 7.029−7.082 ppm (m, 8H); $^{13}$C NMR (TMS, CDCl$_3$, 101 MHz): $\delta = 139.645$, 139.617, 135.774, 133.668, 133.450, 132.059, 131.807, 125.666, 125.617, 123.674, 123.640, 121.660, 120.376, 120.249, 120.112, 109.638, 109.585 ppm; LDI-TOF: m/z (%): 487.0792 (100) [M$^+$].

**Synthesis of 9,9'-(4,5-Dibromo-1,2-phenylene)bis(9H-carbazole) (DCzDBr).** A mixture of carbazole (3.60 g, 21.5 mmol), 1,2-dibromo-4,5-difluorobenzene (2.72 g, 10 mmol), and K$_2$CO$_3$ (3.34 g, 24 mmol) in dimethyl sulfoxide (DMSO) (30 mL) was stirred at 150 °C for 12 h under argon atmosphere. After cooling to room temperature, the mixture was poured into water and then filtered. The crude product was dried and then purified by column chromatography and recrystallization from ethyl acetate to afford white solid with a yield of 75% (4.25 g). $^1$H NMR (TMS, CDCl$_3$, 400 MHz): $\delta = 8.090$ (s, 2H), 7.773-7.795 (m, 4H), 7.126-7.149 (m, 4H), 7.045-7.089 ppm (m, 8H); $^{13}$C NMR (TMS, CDCl$_3$, 101 MHz): $\delta = 139.401$, 134.897, 134.550, 125.773, 124.427, 123.735, 120.560, 120.170, 109.468 ppm; LDI-TOF: m/z (%): 565.9821 (100) [M$^+$].

**Synthesis of 9,9'-(4-Bromo-5-(diphenylphosphaneyl)-1,2-phenylene)bis(9H-carbazole) (DCzSBrSP).** To a solution of 9,9'-(4,5-dibromo-1,2-phenylene)bis(9H-carbazole) (5.66 g, 10 mmol) in mixed Et$_2$O (25 mL) and tetrahydrofuran (THF, 60 mL) was added with n-BuLi (2.5 M in hexane, 4 mL, 10 mmol) at -120 °C. Then, the mixture was stirred for 30 min at the same temperature. The mixture was added with Ph$_2$PCl (1.8 mL, 10 mmol) during 30 min, and then the solution naturally returned to room temperature. The mixture was treated with aqueous NH$_4$Cl and extracted with CH$_2$Cl$_2$. The organic phase was separated and dried over anhydrous Na$_2$SO$_4$. After removal of the solvents, the residue was purified by column chromatography and recrystallization with ethyl acetate to give white solid with a yield of 64% (4.30 g). $^1$H NMR (TMS, CDCl$_3$, 400 MHz): $\delta = 8.078$ (d, $J = 3.2$ Hz, 1H), 7.758-7.779 (m, 2H), 7.700 (d, $J = 7.6$ Hz, 2H), 7.336−7.462 (m, 10H), 7.232 (d, $J = 1.6$ Hz, 1H), 7.113−7.135 (m, 2H), 6.893−7.065 ppm (m, 10H); $^{13}$C NMR (TMS, CDCl$_3$, 101 MHz): $\delta = 140.369$, 140.208, 139.248, 139.060, 135.916, 135.061, 135.019, 134.959, 134.131, 134.106, 134.060, 133.857, 133.085, 129.586, 129.093, 129.021, 125.618, 125.383, 123.687, 123.469, 120.389, 120.167, 120.003, 119.824, 109.566, 109.375 ppm; $^{31}$P NMR (TMS, CDCl$_3$, 162 MHz): $\delta = -4.629$ ppm; LDI-TOF: m/z (%): 673.1232 (100) [M$^+$].

**Synthesis of (2-bromo-4,5-di(9H-carbazol-9-yl)phenyl)diphenyl-phosphine oxide (DCzSBrSPO).** To a solution of DCzSBrSP (1.34 g, 2 mmol) in dichloromethane (DCM, 10 mL) was added with 30% $H_2O_2$ at 0 °C. The mixture was further stirred for 4 h at the room temperature. The mixture was washed with aq. $NaHSO_3$, and then extracted with $CH_2Cl_2$. The extracts were dried over anhydrous $Na_2SO_4$. After removing DCM, the residue was purified with column chromatography to give white solid with a yield of ~100% (1.37 g). $^1$H NMR (TMS, CDCl$_3$, 400 MHz): $\delta$ = 8.184 (d, $J$ = 2.8 Hz, 1H), 7.855–7.901 (m, 4H), 7.699–7.782 (m, 5H), 7.540–7.555 (m, 6H), 6.948–7.121 ppm (m, 12H); $^{13}$C NMR (TMS, CDCl$_3$, 101 MHz): $\delta$ = 137.812, 137.778, 136.861, 136.832, 136.418, 136.304, 135.125, 135.041, 133.133, 132.116, 131.641, 131.572, 131.508, 131.475, 131.449, 130.995, 130.940, 130.897, 130.374, 129.295, 127.935, 127.810, 124.733, 124.536, 124.164, 124.114, 122.810, 122.550, 119.732, 119.430, 119.031, 118.926, 108.365, 108.025 ppm; $^{31}$P NMR (TMS, CDCl$_3$, 162 MHz): $\delta$ = 30.184 ppm; LDI-TOF: m/z (%): 689.1196 (100) [M$^+$].

The original NMR, high-resolution MS (HRMS) and high-performance liquid chromography (HPLC) spectra were provided in Supplementary Note 6 (Supplementary Figs. 22–36).

## Gaussian simulation

Theoretical computations were carried out on the basis of the restricted and unrestricted formalism of Beck's three-parameter hybrid exchange functional[47] and Lee, and Yang and Parr correlation functional[48] (B3LYP). The optimization was also performed at the level of 6-31 G(d,p), respectively, on the basis of single-crystal data. The fully optimized stationary points were further characterized by harmonic vibrational frequency analysis to ensure that real local minima had been found without imaginary vibrational frequency. The total energies were also corrected by zero-point energy both for the ground and excited states. Natural transition orbital (NTO) analysis was performed on the basis of optimized ground-state geometries at the same level[49].

## Data availability

The authors declare that the data generated in this study are provided in Supplementary Information. Crystallographic data for the structures reported in this Article have been deposited at the Cambridge Crystallographic Data Centre, under deposition numbers 2309045 (DCzSBr) [https://doi.org/10.5517/ccdc.csd.cc2hhrb3], 2309044 (DCzDBr) [https://doi.org/10.5517/ccdc.csd.cc2hhr92], 2309047 (DCzSBrSP) [https://doi.org/10.5517/ccdc.csd.cc2hhrd5] and 2309046 (DCzSBrSPO) [https://doi.org/10.5517/ccdc.csd.cc2hhrc4], respectively. Copies of the data can be obtained free of charge via https://www.ccdc.cam.ac.uk/structures/.

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

## Acknowledgements

H.X. thanks Prof. Runfeng Chen for his constructive suggestions on DRTP mechanism. The authors thank the support by the National Natural Science Foundation of China (22325502, 92061205, 62175060, 52273173, 22005088).

## Author contributions

H.X. conceived the projects. G.L., J.T., H.W., Y.M., and C.D. performed the experiments. H.X., G.L., Y.M., S.C., J.Z., and C.H. analyzed the data and wrote the paper. All authors commented on the manuscript.

## Competing interests

The authors declare no competing interests.
