## [Peer Review File · Nature Communications]

Delayed Room Temperature Phosphorescence Enabled by PhosphinesReviewer #1 (Remarks to the Author):

In this work, the authors reported delayed room temperature phosphorescence RTP from carbazole-phosphine hybrids named DCzSBrSP and DCzSBrSPO, through rationally regulating the mixed n-n and p-n interactions. Two carbazole groups at ortho positions can effectively enhance the stability of $3(n, n^*)$ states, and diphenylphosphine and diphenylphosphine oxide with $3(n, n^*)$ can be used as medium groups to induce delayed RTP. Compared with the instantaneous RTP molecules DCzSBr and DCzDBr, DCzSBrSP and DCzSBrSPO not only achieved the lifetimes increased by 9 and 6 times, but also realized RTP emission with a delay of tens of milliseconds. The authors used time-resolved technology to demonstrate that different to DCzSBr and DCzDBr, most of T1 states for DCzSBrSP and DCzSBrSPO transitioned to their phosphine-based TM states, and then transitioned to their carbazole-based Tn^* states after tens of milliseconds, therefore generating delayed RTP. Then, the authors proved this delayed RTP phenomenon can be used for incremental information displays. These results are very interesting, and the integrity of this work was satisfying. This work can improve the theory of organic afterglow. So, I would like to recommend the publication of this work in Nature Communications after some minor revisions:

1. In Figure 2b: why were DCzDBr and DCzSBr sensitive to excitation wavelengths at 350nm and 390nm, but DCzSBrSP and DCzSBrSPO did not show strong excitation dependence?
2. In the supporting information, in Figure S4: the lifetime decay curve of DCzSBrSPO at 502 nm revealed a long tail. The authors should provide the detailed measurement condition or give a rational explanation.
3. The preparation process for information encryption and anti-counterfeiting applications should be described in detail in the supporting information.
4. Besides oxidation, phosphine molecules can be converted to phosphine sulfide molecules. S atom can also form p-n interactions with adjacent carbazole groups. So, I wonder P=S group can also generate the similar delayed RTP or not?
5. There are some typos should be corrected. For example, on page 4, in the third line of the second paragraph, the numbers 9 and 6 folds were reversed.

Reviewer #2 (Remarks to the Author):

The authors reported that carbazole-substituted bromotriphenylphosphine derivatives had a new phenomenon "delayed room temperature phosphorescence". According to delayed room-temperature phosphorescence properties, a series of photophysical characterizations and theoretical calculations, the authors claimed that phosphine groups provided the $3(n, n^*)$ intermediate state (TM) between $3(n, n^*)$ -featured T1 and Tn^* states to form a gradual transition process, which the transition from TM to Tn^* is a long-term process, so a delayed room temperature phosphorescence was observed. This is the first example of delayed room-temperature phosphorescence. The "incremental" information encryption in the practical application part is interesting. So, I recommend the publication of this work in Nature Communications, but there are some minor issues that need to be revised or improved before publication.

1. In Figure 1c, the pictures of DCzDBr before and after turning off the UV lamp are absent, despite the legends containing DCzDBr. Please add here or in the support information section.
2. In caption of Figure 2, the order of DCzDBr and DCzSBr does not match the figures and seems to be reversed. The Figure 3 seems to have the same problem. Please recheck it.
3. In the conclusion of the last paragraph, there is typos, such as different, immediately. Please recheck the text.
4. DCzSBrSPO seems to exhibit better delayed room temperature phosphorescence behavior. Is DCzSBrSPO more suitable for information encryption and anti-counterfeiting applications than DCzSBrSP? Why did the author choose DCzSBrSP in the practical application section instead of DCzSBrSPO?
5. Figure 4a of the application section, after turning off the excitation light source, the Chinese and English parts composed of DCzSBrSP seem to emit stronger light in the middle than on both sides, why is that?
6. Is this strategy of phosphine mediated delayed room temperature phosphorescence universally applicable? Will introducing phosphine into arylcarbazole derivatives necessarily achieve this

delayed room temperature phosphorescence phenomenon?

Reviewer #3 (Remarks to the Author):

This manuscript described a simple but effective strategy to achieve a brand-new delayed room temperature phosphorescence phenomenon. The authors demonstrated that the phosphine groups introduced into carbazole derivatives can construct intermediate triplet level (T_m) and realize new long-term energy transfer pathways, thereby achieving delayed room temperature phosphorescence. Investigations based on time-resolved emission spectra, intermolecular interactions and theoretical calculations indicated the critical role of phosphorus atom in this delayed process. As a result, the organophosphorus molecules achieved several folds increased lifetimes and time delays for tens of milliseconds. This interesting delayed organic afterglow phenomenon was reported for the first time, and the roles of phosphorus atoms in the emission process suggested a universal and feasible strategy for realizing this unique property. So, I recommend acceptance of this manuscript for publication.

Minor revisions are suggested:

1. Even for the same materials, why the luminescence duration times of the materials in Figure 1c were longer than the materials in Figure 4?
2. DCzDBr can be emissive for a longer time in Figure 4a than in Figure 4b (e.g., at 0.5s), please give a reasonable explanation.
3. Did TPPBr molecule also have the property of room temperature phosphorescence?
4. It is known that phosphine can be oxidized into phosphine oxide. I wonder under UV excitation, the phosphine molecules can be stable enough or not to make the delayed afterglow repeatable?

Responses to Reviewers' Comments

Reviewer #1:

In this work, the authors reported delayed room temperature phosphorescence RTP from carbazole-phosphine hybrids named DCzSBrSP and DCzSBrSPO, through rationally regulating the mixed π - π and p- π interactions. Two carbazole groups at ortho positions can effectively enhance the stability of $^3(\pi, \pi^*)$ states, and diphenylphosphine and diphenylphosphine oxide with $^3(n, \pi^*)$ can be used as medium groups to induce delayed RTP. Compared with the instantaneous RTP molecules DCzSBr and DCzDBr, DCzSBrSP and DCzSBrSPO not only achieved the lifetimes increased by 9 and 6 times, but also realized RTP emission with a delay of tens of milliseconds. The authors used time-resolved technology to demonstrate that different to DCzSBr and DCzDBr, most of T_1 states for DCzSBrSP and DCzSBrSPO transited to their phosphine-based T_M states, and then transited to their carbazole-based T_n^* states after tens of milliseconds, therefore generating delayed RTP. Then, the authors proved this delayed RTP phenomenon can be used for incremental information displays. These results are very interesting, and the integrity of this work was satisfying. This work can improve the theory of organic afterglow. So, I would like to recommend the publication of this work in Nature Communications after some minor revisions.

Response: Reviewer's kind approval and accurate summary of our work is highly appreciated. Thanks a lot!

1. In Figure 2b: why were DCzDBr and DCzSBr sensitive to excitation wavelengths at 350 nm and 390 nm, but DCzSBrSP and DCzSBrSPO did not show strong excitation dependence?

Response: Thanks a lot for this constructive comment! We should apologize for our unclear description inducing this misunderstanding. As shown in absorption spectra of these four compounds in Figure S1, DCzSBr displays two absorption peaks at about 350 nm and 390 nm, as the same as DCzDBr, despite the weaker absorption bands of the latter. These two absorption bands can be attributed to $S_0 \rightarrow T_1$ transitions of DCzSBr and DCzDBr powders with stabilized carbazole groups, which are mirroring symmetric with their single-molecular phosphorescence spectra, in accord with Franck-Condon Principle. As a result, in Figure 2b, phosphorescence emissions of DCzSBr and DCzDBr reveal strong correlations with the excitation bands near 350 nm and 390 nm. In contrast, aryl phosphine groups in DCzSBrSP and DCzSBrSPO introduce additional absorption bands in this region, which overlap with the triplet absorptions of their carbazole groups. Therefore, emission-excitation mapping of DCzSBrSP and DCzSBrSPO show

broad contours rather than relatively concentrated patterns like the situations of DCzSBr. Figure S1 was revised to highlight the triplet absorption bands of DCzSBr and DCzDBr, and the related discussion was added. Thanks a lot!

Revision:

Figure S1 was revised, and the related discussion was added:

Figure S1. Electronic absorption spectra of DCzSBr, DCzDBr, DCzSBrSP and DCzSBrSPO powders. The dotted lines highlight the absorption bands of DCzSBr and DCzDBr in the range of 350-400 nm.

“DCzSBr and DCzDBr reveal two absorption peaks in the range of 350-400 nm, which can be attributed to $S_0 \rightarrow T_1$ transitions of stabilized carbazole groups in DCzSBr and DCzDBr powders, according to Franck-Condon Principle (Figure S1). Therefore, it is rational that phosphorescence emissions of DCzSBr and DCzDBr have strong correlations with these excitation bands (Figure 2b). In contrast, absorption bands of aryl phosphine groups in DCzSBrSP and DCzSBrSPO also locate in this region, which overlap with the triplet absorptions of their carbazole groups. As consequence, their emission-excitation mapping shows broad and structure-less contours in this region.”

2. In the supporting information, in Figure S4: the lifetime decay curve of DCzSBrSPO at 502 nm revealed a long tail. The authors should provide the detailed measurement condition or give a rational explanation.

Response: Thanks a lot for this constructive comment! The tail in nanosecond-scale time decay curve of DCzSBrSPO at 502 nm can be attributed to its single-molecular phosphorescence with the lifetime at microsecond scale. To make it clear, microsecond time decay of DCzSBrSPO at 502 nm were measured and separately shown in Figure S9, which is compared with single-molecular phosphorescence decays of DCzSBr, DCzDBr and DCzSBrSP in Figure S8. Obviously, phosphorescence of DCzSBrSPO powder has two components with lifetimes of 0.3 and 3.15 ms, respectively. The shorter one is involved in nanosecond time decay as a tail. Figures S8 and S9 and related discussions were added in the revision. Thanks a lot!

Revision:

The microsecond time decay curve of DCzSBrSPO powder at 502nm and related discussions were added:

“**Figure S6.** Lifetime decays of fluorescent components from DCzSBr, DCzDBr, DCzSBrSP and DCzSBrSPO powders at nanosecond range. The tail of DCzSBrSPO at 502 nm can be attributed to its single-molecular phosphorescence at microsecond scale as shown in Figure S7.”

Figure S8. Lifetime decays of single-molecular phosphorescence from DCzSBr, DCzDBr and DCzSBrSP powders at microsecond range.

Figure S9. Lifetime decay of single-molecular phosphorescence from DCzSBrSPO powder at millisecond range, which consists of two phosphorescence components with lifetimes of 0.3 and 3.15 ms, respectively.

3. The preparation process for information encryption and anti-counterfeiting applications should be described in detail in the supporting information.

Response: Thanks a lot for this constructive comment! We should apologize for the insufficient information. The detailed descriptions of pattern preparation for information encryption and anti-counterfeiting applications were added in the third part “Pattern Preparation for Applications” of Experimental section of supporting information. Thanks a lot!

Revision:

Detailed descriptions of pattern preparation was added:

3. Pattern Preparation for Applications

3.1 Material combination

For multilevel information displays, materials featuring strong fluorescence and instant and delayed RTP with different lifetimes, namely DMAC-DPS, DCzSBr, DCzDBr and DCzSBrSP, were chosen for information encryption and anti-counterfeiting applications. DMAC-DPS is bis[4-(9,9-dimethyl-9,10-dihydroacridine)phenyl]sulfone with thermally activated delayed fluorescence at microsecond level chosen

as emission background. Compared to another delayed RTP material DCzSBrSPO, RTP duration of DCzSBrSP is longer, which provides more time for distinguishing complicated information. More importantly, RTP intensity of DCzSBrSP is comparable to those of DCzSBr and DCzDBr, making information encryption mainly dependent on emission duration.

Scheme S2. Flow diagram of printing the password panel for non-linear time encoded security application. Black arrows indicate the procedure.

3.2 General Procedure for Pattern Preparation

- i. *Platemaking*: screens used for printing were customized according to the separated elements of the patterns (Figure 4a and Scheme S2).
- ii. *Ink preparation*: all the materials were ground and then sieved. The inks were prepared through uniformly dispersing the powders in aloe vera gel. Inks of DCzDBr, DCzSBrSP and DCzSBr were directly used to print the patterns in Figure 4a; while, for Scheme 2 and Figure 4b, Inks of DCzDBr, DCzSBrSP and DCzSBr were mixed with DMAC-DPS based ink, so that under UV excitation, RTP can be covered by blue fluorescence from DMAC-DPS.

iii. *Printing*: The separated elements of the patterns were sequentially printed with the corresponding inks through screens prepared in step (i). For the pattern in Figure 4a, DCzDBr based ink was firstly used to print the “main building of Heilongjiang University” on filter paper. After this pattern dried, DCzSBrSP based ink was used to print the university name in Chinese and English. Finally, DCzSBr based ink was used to print the year numbers. The Chinese characters of “周年” were used to calibrate the positions of all the elements. For the patterns in Figure 4b, as shown in Scheme S2, four screens were used to the patterns on the filter paper through four steps and the corresponding inks.

4. Besides oxidation, phosphine molecules can be converted to phosphine sulfide molecules. S atom can also form p- π interactions with adjacent carbazole groups. So, I wonder P=S group can also generate the similar delayed RTP or not?

Response: Thanks a lot for this constructive comment! According to reviewer’s suggestion, we synthesized the sulfide of DCzSBrSP named DCzSBrSPS, and performed structure characterization and emission property measurement. It is shown that DCzSBrSPS does not display the long-persistence RTP, which can not be observed with naked eyes, since its phosphorescent lifetime is less than 20ms. The synthesis procedure and structural characterization of DCzSBrSPS were added in the experimental section of supporting information, and its phosphorescence spectra and time decay curves were added as Figure S2 and S3. The related discussion was also added in the revision. Thanks a lot!

Revision:

The synthesis and characterization of DCzSBrSPS were added:

“However, sulfide of DCzSBrSP dose not exhibit long-persistence room temperature phosphorescence, since its lifetimes of its powder are less than 20ms (Figs S2 and S3).”

“(2-Bromo-4,5-di(9H-carbazol-9-yl)phenyl)diphenylphosphine sulfide (DCzSBrSPS): A mixture of DCzSBrSP (1.34g, 2 mmol) and S₈ (2.56 g, 1 mmol) in chloroform (10 mL) was stirred at 60°C for 10 h. After removing chloroform, the residue was purified with column chromatography to give white solid with a yield of ~100%. ¹H NMR (TMS, CDCl₃, 400 MHz): δ = 8.188 (d, *J* =3.6 Hz, 1H), 7.990-8.030 (m, 4H), 7.774 (d, *J* =6.8 Hz, 2H), 7.719 (d, *J* =6.4 Hz, 2H), 7.607 (d, *J* =13.6 Hz, 1H), 7.485-7.572 (m, 6H), 6.937-7.133 ppm (m, 12H); ¹³C NMR (TMS, CDCl₃, 100 MHz): δ =138.879, 138.767, 137.273, 137.245, 136.759, 136.677, 136.626, 136.505, 134.283, 133.429, 132.692, 132.560, 132.359, 132.252, 132.208, 132.180,

131.449, 130.575, 129.069, 128.941, 125.831, 125.598, 125.128, 125.068, 123.919, 123.661, 120.841, 120.572, 120.142, 120.018, 109.517, 109.205 ppm; ^{31}P NMR (TMS, CDCl_3 , 121.5 MHz): $\delta = 45.798$ ppm.”

Scheme S1. v. S_8 , CHCl_3 , 60°C , 10h.

Figure S2. Steady-state photoluminescence (PL, black line) and time-resolved RTP (red line) spectra of DCzSBrSPS powder under ambient condition and excitation at 330 nm.

Figure S3. Time decays of DCzSBrSPs powder at 562nm and 610nm, respectively, under excitation at 360nm.

5. There are some typos should be corrected. For example, on page 4, in the third line of the second paragraph, the numbers 9 and 6 folds were reversed.

Response: Thanks a lot for this kind reminding! We should apologize for our carelessness. The mistakes were corrected in the revision. Thanks a lot!

Revision:

Inverted numbers are corrected:

“For RTP peaks at 540 nm excited by 390 nm, compared to DCzSBr and DCzDBr with RTP lifetimes of 116 and 80 milliseconds, DCzSBrSP exhibits the longest RTP lifetime at 540 nm reaching 742 milliseconds, which is elongated by more than 6 and 9 folds, respectively (Fig. 2c).”

Reviewer #2:

The authors reported that carbazole-substituted bromotriphenylphosphine derivatives had a new phenomenon “delayed room temperature phosphorescence”. According to delayed room-temperature phosphorescence properties, a series of photophysical characterizations and theoretical calculations, the authors claimed that phosphine groups provided the $^3(n, \pi^*)$ intermediate state (T_M) between $^3(\pi, \pi^*)$ -featured T_1 and T_n^* states to form a gradual transition process, which the transition from T_M to T_n^* is a long-term process, so a delayed room temperature phosphorescence was observed. This is the first example of delayed room-temperature phosphorescence. The “incremental” information encryption in the practical application part is interesting. So, I recommend the publication of this work in Nature Communications, but there are some minor issues that need to be revised or improved before publication.

Response: We are grateful to reviewer for the kind approval and accurate summary of our work. Thanks a lot!

1. In Figure 1c, the pictures of DCzDBr before and after turning off the UV lamp are absent, despite the legends containing DCzDBr. Please add here or in the support information section.

Response: Thanks a lot for this constructive comment! We should apologize for our carelessness. Photos of emission process for DCzDBr powder was added in Figure 1c. Thanks a lot!

Revision:

Photos of DCzDBr were added in Figure 1c:

Figure 1. Molecular Design of binary phosphine-carbazole systems for delayed room temperature phosphorescence (DRTP). (c) Photos of DCzSBr, DCzDBr, DCzSBrSP and DCzSBrSPO powders excited with UV light at 365 nm and after UV excitation for 1.6-4.0 seconds.

2. In caption of Figure 2, the order of DCzDBr and DCzSBr does not match the figures and seems to be reversed. The Figure 3 seems to have the same problem. Please recheck it.

Response: Thanks a lot for this constructive comment! We should apologize for our carelessness inducing these mistakes. We rechecked and corrected the mistakes in the captions of Figures 2 and 3. Thanks a lot!

Revision:

Captions of Figures 2 and 3 were corrected:

“Figure 2. Photophysical properties of RTP molecules. (a) Steady-state photoluminescence (PL, dashed lines) and time-resolved RTP (solid lines) spectra of DCzSBr, DCzDBr, DCzSBrSP and DCzSBrSPO powders under ambient condition. RTP spectra were recorded after a delay of 30 ms. For DCzSBr, DCzDBr and DCzSBrSP, two excitation wavelengths of 350 and 350 nm are chosen to indicate the excitation dependence of their PL and RTP emissions. (b) Excitation-emission mapping of DCzSBr, DCzDBr, DCzSBrSP and DCzSBrSPO powders under ambient conditions. (c) Time decays of emission peaks at 590 (above) and 540 nm (below) for DCzSBr, DCzDBr, DCzSBrSP and DCzSBrSPO powders excited by 350 nm.”

“Figure 3. Delayed RTP mechanism of carbazole-phosphine hybrids. (a) Sliced time-resolved emission spectra (TRES) of DCzSBr, DCzDBr, DCzSBrSP and DCzSBrSPO powders. The corresponding energy transfer processes are highlighted as insets.”

3. In the conclusion of the last paragraph, there is typos, such as different, immediately. Please recheck the text.

Response: Thanks a lot for this constructive comment! We apologize again for our carelessness during manuscript preparation. The manuscript was carefully checked again to get rid of the typos and grammatical errors. Thanks a lot!

Revision:

The typos was corrected:

“Different to common instant RTP immediately occurred after excitation removal, delayed RTP can only be recognized with a postponement of dozens of milliseconds, and then gradually enhanced in the next hundreds of milliseconds.”

4. DCzSBrSPO seems to exhibit better delayed room temperature phosphorescence behavior. Is DCzSBrSPO more suitable for information encryption and anti-counterfeiting applications than DCzSBrSP? Why did the author choose DCzSBrSP in the practical application section instead of DCzSBrSPO?

Response: Thanks a lot for this kind suggestion! We should apologize for our insufficient explanations on material selection for the applications. As reviewer pointed out, delayed time of RTP from DCzSBrSPO is longer than that of DCzSBrSP. However, for practical applications, DCzSBrSP is more suitable, since the intensity and duration of its RTP are larger than those of DCzSBrSPO. Especially, the RTP intensity of DCzSBrSP is comparable to those of DCzSBr and DCzDBr. In this case, at the beginning after excitation turn-off, the encrypted information can only be distinguished by their delayed or instant RTP characteristics, namely time resolution, rather than their different RTP intensities. Furthermore, the delayed time of RTP from DCzSBrSP is about 30 ms, which is close to the limit for naked eyes, and undoubtedly increases the difficulty of being cracked. So, we think DCzSBrSP would be more suitable for the applications through combining with the instant RTP analogs. The corresponding explanation was added in the experimental section of supporting information. Thanks a lot!

Revision:

Explanation on material combination for applications was added:

“3.1 Material combination

For multilevel information displays, materials featuring strong fluorescence and instant and delayed RTP with different lifetimes, namely DMAC-DPS, DCzSBr, DCzDBr and DCzSBrSP, were chosen for information encryption and anti-counterfeiting applications. DMAC-DPS is bis[4-(9,9-dimethyl-9,10-dihydroacridine)phenyl]sulfone with thermally activated delayed fluorescence at microsecond level chosen as emission background. Compared to another delayed RTP material DCzSBrSPO, RTP duration of DCzSBrSP is longer, which provides more time for distinguishing complicated information. More importantly, RTP intensity of DCzSBrSP is comparable to those of DCzSBr and DCzDBr, making information encryption mainly dependent on emission duration.”

5. Figure 4a of the application section, after turning off the excitation light source, the Chinese and English parts composed of DCzSBrSP seem to emit stronger light in the middle than on both sides, why is that?

Response: Thanks a lot for this constructive comment! We apologize again for the insufficient information. We use a circular ultraviolet flashlight as excitation source for the applications, whose spot is round with a diameter of 2 cm when placed 10 cm above the samples. Therefore, the UV light intensity on the Chinese and English characters in the center of sample was the strongest, inducing their strongest RTP emissions. The explanation was added in the experimental section of supporting information. Thanks a lot!

Revision:

Explanation on spotlight parameters added:

“For anti-counterfeiting and encryption applications, a circular ultraviolet flashlight (GET-104) was used as the excitation source with a power of 3 W and the peak wavelength at 365 nm, whose spot diameter is about 2 cm at the distance of 10 cm above the samples.”

6. Is this strategy of phosphine mediated delayed room temperature phosphorescence universally applicable? Will introducing phosphine into arylcarbazole derivatives necessarily achieve this delayed room temperature phosphorescence phenomenon?

Response: Thanks a lot for this constructive comment! Although our work reports the first delayed RTP phenomenon can be distinguished by naked eyes, this phenomenon should be universally applicable in term of the “intermediate triplet energy level” mechanism. Actually, in our work, it is already indicated that not only P atom but also O atom can be used to generate $^3n\pi^*$ states as the intermediate triplet levels. In this sense, all the heteroatom-containing groups with suitable intermolecular n- π interactions have the potential for constructing delayed RTP materials. Thanks a lot!

Reviewer #3:

This manuscript described a simple but effective strategy to achieve a brand-new delayed room temperature phosphorescence phenomenon. The authors demonstrated that the phosphine groups introduced into carbazole derivatives can construct intermediate triplet level (T_m) and realize new long-term energy transfer pathways, thereby achieving delayed room temperature phosphorescence. Investigations based on time-resolved emission spectra, intermolecular interactions and theoretical calculations indicated the critical role of phosphorus atom in this delayed process. As a result, the organophosphorus molecules achieved several folds increased lifetimes and time delays for tens of milliseconds. This interesting delayed organic afterglow phenomenon was reported for the first time, and the roles of phosphorus atoms in the emission process suggested a universal and feasible strategy for realizing this unique property. So, I recommend acceptance of this manuscript for publication.

Response: We highly appreciate reviewer's pertinent evaluation of our work! Thanks a lot!

1. Even for the same materials, why the luminescence duration times of the materials in Figure 1c were longer than the materials in Figure 4?

Response: Thanks a lot for this constructive comment! We should apologize for our unclear expressions. Figure 1c shows the photos of as-prepared pure powders, while the patterns in Figure 4 were prepared with grounded powders dispersed in aloe vera gel. Therefore, not only the molecular packing, but also concentrations of the materials used in Figure 4 were reduced, leading to the weaker RTP intensities. This situation is in accord with previous reports (*Angew. Chem. Int. Ed.* **2021**, *60*, 9500-9506; *Angew. Chem. Int. Ed.* **2018**, *57* (51), 16821-16826). Therefore, under the same excitation intensities, the weaker initial RTP intensities in Figure 4 inevitably induced the shorter duration for naked eyes. The material preparation details for Figure 4 were added in the experimental section of supporting information. Thanks a lot!

Revision:**Material preparation details for the applications were added:**

“ii. *Ink preparation:* all the materials were ground and then sieved. The inks were prepared through uniformly dispersing the powders in aloe vera gel.”

2. DCzDBr can be emissive for a longer time in Figure 4a than in Figure 4b (e.g., at 0.5s), please give a reasonable explanation.

Response: Thanks a lot for this constructive comment! As reviewer pointed out, actually, not only RTP of DCzDBr based patterns but also RTP from the other patterns became shorter. It is because the patterns in Figure 4a were printed with RTP materials based inks directly, but the patterns in Figure 4b were printed with mixed inks of RTP materials and DMAC-DPS. Therefore, the concentrations of RTP materials in the mixed inks were lower, rendering the weaker RTP intensities and thereby shorter duration for naked eyes. The descriptions of the ink contents were added in the experimental section of supporting information for clarity. Thanks a lot!

Revision:

Material preparation details for the applications were added:

“Inks of DCzDBr, DCzSBrSP and DCzSBr were directly used to print the patterns in Figure 4a; while, for Scheme 2 and Figure 4b, Inks of DCzDBr, DCzSBrSP and DCzSBr were mixed with DMAC-DPS based ink, so that under UV excitation, RTP can be covered by blue fluorescence from DMAC-DPS.”

3. Did TPPBr molecule also have the property of room temperature phosphorescence?

Response: Thanks a lot for this constructive comment! Accordingly, we measured time decay of TPPBr powder with emission peak at 490nm, corresponding to room temperature phosphorescence with shorter lifetimes of several microseconds (Figure C1). Therefore, it is rational that TPPBr can provide a stable intermediate triplet state for delayed RTP. Thanks a lot!

Figure C1. Time decay curve of phosphorescence from TPPBr powder peaked at 490nm.

4. It is known that phosphine can be oxidized into phosphine oxide. I wonder under UV excitation, the phosphine molecules can be stable enough or not to make the delayed afterglow repeatable?

Response: Thanks a lot for this constructive comment! According to our experience, phosphine in solution can be easily oxidized by solved oxygen. However, in solid states, phosphine can be very stable. We further tested the photostability of DCzSBrSP powder under UV irradiation for 72 h (Figure C2). It is shown that its RTP intensities were nearly unchanged. To figure out the repeatability of delayed RTP from DCzSBrSP, we repeated the password panel application as the same as Figure 4b. It is shown that even after 100 times of UV excitation, the evolution of patterns was unchanged (Figure C2). So, it is convincing that DCzSBrSP is stable enough under the common irradiation and excitation conditions for the practical applications. Thanks a lot!

Figure C2. RTP intensity variation of **DCzSBrSP** powder exposed to UV irradiation at 365 nm from an ultraviolet flashlight (3w) for 72 hours.

Figure C3. Incremental afterglow display from a password panel using DCzSBrSP as delayed RTP material, after 100 times of UV excitation with a 3W 365 nm UV flashlight.

Reviewer #1 (Remarks to the Author):

The authors have addressed all the concerns, the revised manuscript can be accepted for publication in Nature Communications now.

Reviewer #2 (Remarks to the Author):

This version can be accepted.

Reviewer #3 (Remarks to the Author):

The authors have made appropriate revisions in response to the reviewers' comments. The manuscript is now high-quality work and can be accepted in the current version.